# Collective forces of tumor spheroids in three-dimensional biopolymer networks

Christoph Mark[1‡*], Thomas J Grundy[2,3‡], Pamela L Strissel[4‡], David Böhringer[1‡], Nadine Grummel[1], Richard Gerum[1], Julian Steinwachs[1], Carolin C Hack[4], Matthias W Beckmann[4], Markus Eckstein[5], Reiner Strick[4†], Geraldine M O'Neill[2,3†], Ben Fabry[1†*]

[1]Department of Physics, Friedrich-Alexander University Erlangen-Nürnberg, Erlangen, Germany; [2]Children's Cancer Research Unit, The Children's Hospital at Westmead, Sydney, Australia; [3]School of Medical Sciences and Children's Hospital at Westmead Clinical School, University of Sydney, Sydney, Australia; [4]Department of Gynecology and Obstetrics, Laboratory for Molecular Medicine, University Hospital Erlangen, Friedrich-Alexander University Erlangen-Nürnberg, Erlangen, Germany; [5]Institute of Pathology, University Hospital Erlangen, Erlangen, Germany

**Abstract** We describe a method for quantifying the contractile forces that tumor spheroids collectively exert on highly nonlinear three-dimensional collagen networks. While three-dimensional traction force microscopy for single cells in a nonlinear matrix is computationally complex due to the variable cell shape, here we exploit the spherical symmetry of tumor spheroids to derive a scale-invariant relationship between spheroid contractility and the surrounding matrix deformations. This relationship allows us to directly translate the magnitude of matrix deformations to the total contractility of arbitrarily sized spheroids. We show that our method is accurate up to strains of 50% and remains valid even for irregularly shaped tissue samples when considering only the deformations in the far field. Finally, we demonstrate that collective forces of tumor spheroids reflect the contractility of individual cells for up to 1 hr after seeding, while collective forces on longer timescales are guided by mechanical feedback from the extracellular matrix.

**\*For correspondence:**
christoph.mark@fau.de (CM);
ben.fabry@fau.de (BF)

[†]These authors contributed equally to this work
[‡]These authors also contributed equally to this work

**Competing interests:** The authors declare that no competing interests exist.

## Introduction

In the process of tumor invasion, cancer cells leave the primary tumor either individually or collectively (*Friedl and Wolf, 2003*). This process requires that cells exert physical forces onto the surrounding extracellular matrix (*Friedl and Gilmour, 2009*; *Koch et al., 2012*). As cellular force generation and cell-matrix interactions are increasingly recognized as potential therapeutic targets against cancer cell invasion and metastasis (*Holle et al., 2018*; *Chaudhuri et al., 2018*), there is a need to quantify the forces that are collectively exerted by invading cancer cells under physiologically relevant conditions. In this work, we introduce a computationally and experimentally simple and reliable method that captures collective effects in tissue remodeling and thus facilitates screenings of potential force-targeting agents.

Numerous biophysical assays have been developed to quantify the traction forces of single cancer cells by measuring the deformations that a cell induces in linear elastic substrates (2D and 3D) with known stiffness (*Dembo and Wang, 1999*; *Butler et al., 2002*; *Legant et al., 2010*). This technique has since been extended to multicellular systems to study collective cell guidance by intercellular stresses in 2D cell monolayers (*Tambe et al., 2011*; *Trepat et al., 2009*). Likewise, intercellular stresses within 3D multicellular aggregates (so-called spheroids) have been studied by quantifying the deformation of small elastic beads that are embedded in the spheroids (*Dolega et al., 2017*).

All methods referenced above are based on linear elastic materials that exhibit a constant stiffness, independent of strain, so that the measured deformation is proportional to the corresponding force. To mimic the physiological condition of cells invading connective tissue in vitro, however, cells are typically cultured in non-linear biopolymer networks such as reconstituted collagen that stiffen significantly when extended (*Storm et al., 2005*; *Münster et al., 2013*) but soften when compressed (*Steinwachs et al., 2016*; *Münster et al., 2013*). Considering these nonlinear material properties in a finite element approach allows for the quantification of the total contractility (*Hall et al., 2016*) and the reconstruction of the three-dimensional traction force field around individual cells in a biopolymer network (*Steinwachs et al., 2016*).

Multicellular tumor spheroids embedded in collagen gels are - depending on cell type - able to contract the collagen fiber network, thereby exerting tensile forces in the matrix that in turn realign fiber bundles and facilitate cell invasion into the matrix (*Kopanska et al., 2016*; *Kopanska et al., 2015*; *Chen et al., 2019*; *Han et al., 2016*; *Lee et al., 2017*; *Kaufman et al., 2005*; *Carey et al., 2013*). Thus, multicellular tumor spheroids not only replicate the main structural and functional properties of solid tumors (*Nunes et al., 2019*), but can further serve as a model system for the mechanics of cancer invasion, including collective cellular force generation and tissue remodeling. However, current studies on the force generation of multicellular spheroids all use matrix deformation as a proxy for contractility, to avoid the complex problem of force reconstruction in non-linear materials (*Kopanska et al., 2016*; *Chen et al., 2019*; *Valencia et al., 2015*). This approach poses no problem when comparing spheroids of similar size and cell number. However, in the case of differently sized or differently dense spheroids, or for comparing the collective contractility of a spheroid to that of an individual cell, a more direct measurement in units of force rather than deformation is needed.

Force measurement on a spheroid poses two formidable problems. First, current 3D finite element force reconstruction methods that have been designed for single cells in a non-linear material such as collagen are computationally too slow for analyzing large (~0.5 mm) tumor spheroids (*Steinwachs et al., 2016*). Second, measurements typically require a confocal microscope equipped with a high-resolution (NA 1.0 or higher) water dip-in long working distance objective to image the three-dimensional structure of the collagen fiber network using reflection microscopy (*Steinwachs et al., 2016*). The large scanning volume and associated scanning time would be prohibitive in the case of spheroids.

To overcome these technical challenges, we forgo subcellular force resolution and exploit the approximately spherical symmetry of tumor spheroids. Accordingly, it is sufficient to measure the far-field deformations of the surrounding collagen matrix from a single slice through the equatorial plane of the spheroid, thereby eliminating the need for high-resolution 3D imaging. To quantify matrix deformations over time, image acquisition can be performed with low resolution (4x-10x objective, NA 0.1) brightfield microscopy of micron-sized fiducial markers embedded in the collagen gel.

To relate the measured deformation field surrounding a spheroid to physical forces generated by the cells, we replicate the experiment in silico. Specifically, we simulate a contracting sphere within a bulk of collagen, which can be described by a non-linear material model that takes into account fiber buckling and strain stiffening. We apply this method to spheroids made from glioblastoma cell lines and primary breast cancer cells, as well as to patient-derived breast tumor tissue samples (so-called tumoroids).

## Results

### Collagen contractility assay

We use two model systems to investigate the mechanics of tumor invasion: First, we use in vitro grown tumor spheroids that are generated by culturing suspended cells in non-adhesive U-shaped wells (*Figure 1a*). Second, we use patient-derived tumor tissue samples (tumoroids) with a size of 200–600 μm, similar to the size of the tumor spheroids in our study. Both spheroids and tumoroids are embedded in a 3D collagen matrix by suspending them in an un-polymerized solution of collagen with 1 μm fiducial marker beads (*Figure 1b,c*). After the collagen has polymerized, we track the ongoing cell force-induced deformations of the collagen matrix from brightfield time-lapse images (taken every 5–10 min) using particle image velocimetry (*Taylor et al., 2010*; *Figure 1—figure*

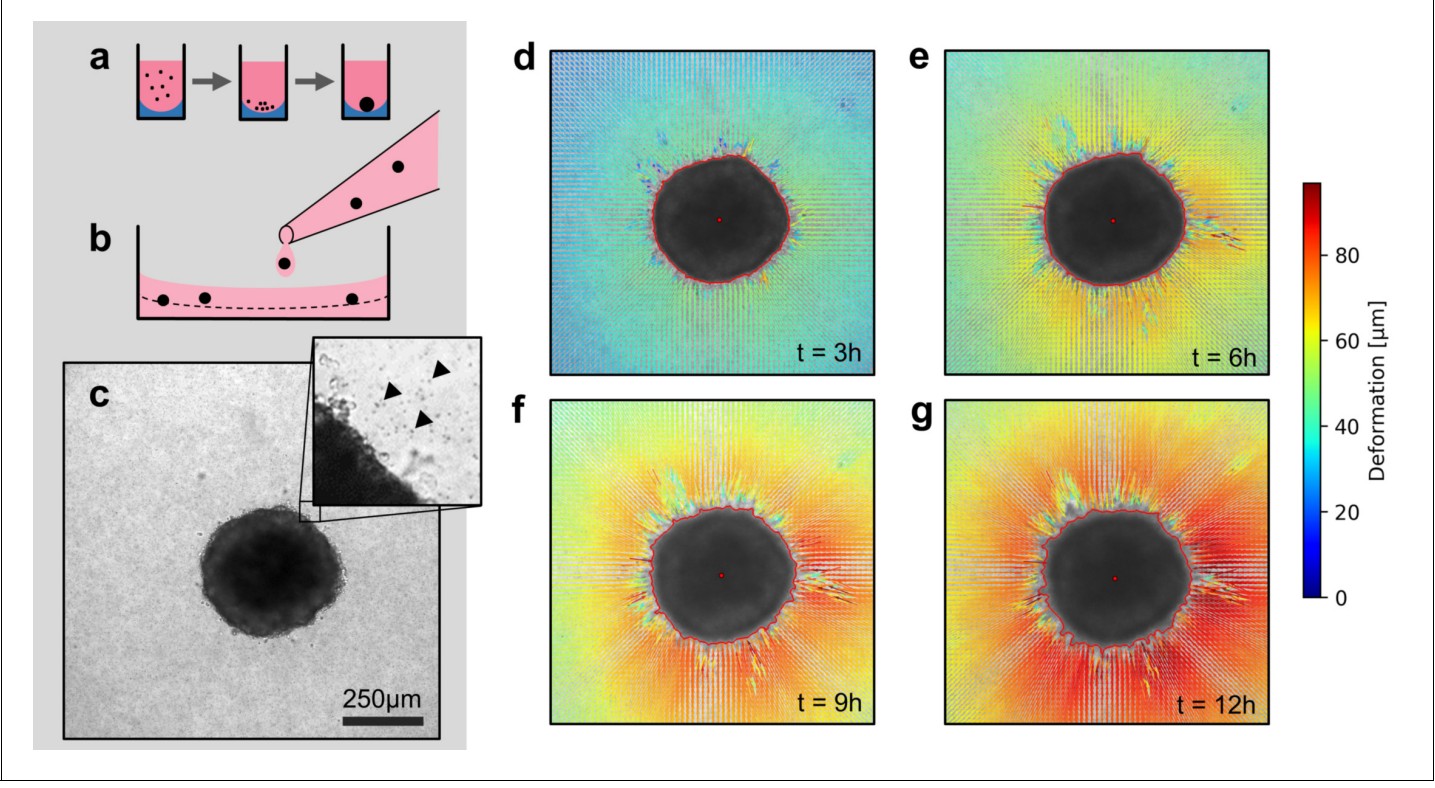

**Figure 1.** Spheroid formation and collagen contractility assay. (**a**) Spheroid generation process within non-adhesive U-shaped wells. (**b**) Spheroid embedding process in collagen gels. The spheroids are suspended in a collagen solution and subsequently pipetted onto a pre-poured layer of collagen (indicated by the dashed line). (**c**) Exemplary brightfield image of the equatorial plane of a U87 spheroid containing 7,500 cells. The inset shows the edge of the tumor spheroid and the micron-sized fiducial markers (arrows) that are added to the collagen solution. (**d-g**) Deformation field obtained by particle image velocimetry, 3 h, 6 h, 9 h and 12 h after the collagen gel has polymerized. The spheroid outline is determined by image segmentation and indicated by the red line.

The online version of this article includes the following figure supplement(s) for figure 1:

**Figure supplement 1.** Displacement during collagen polymerization.
**Figure supplement 2.** Evaluation of Particle Image Velocimetry.
**Figure supplement 3.** Cell proliferation in embedded spheroids.

*supplements 1* and *2*; *Video 1*). In general, we find that both spheroids and tumoroids induce an approximately radially symmetric, inward-directed deformation field with monotonically increasing absolute deformations over time (*Figure 1d–g*; *Videos 2*, *3*, *4*), in line with a previous report on CT26 colon carcinoma cells (*Kopanska et al., 2016*). Cells within the spheroids can proliferate after being embedded in the collagen matrix (*Figure 1—figure supplement 3*). This may lead to spheroid growth and induce a compression of the surrounding matrix (and thus an outward-directed deformation field). However, in none of the spheroids or cell types investigated in this work have we observed such outward-directed matrix deformations.

## Scale-invariant relation between deformation and contractility

To relate the measured deformation field surrounding a spheroid to physical forces generated by the cells, we use the finite element approach described in *Steinwachs et al. (2016)*. Specifically, we simulate a small spherical inclusion with a negative hydrostatic pressure (that emulates contracting cells within the inclusion) within a large surrounding volume of collagen (*Figure 2a,b*). This computational analysis predicts that the absolute deformations of the collagen $u(r)$ are largest directly at the boundary of the inclusion and fall off with increasing distance $r$ from the center, depending on the pressure (*Figure 2b*). For a given pressure, the absolute deformations increase with the radius $r_0$ of the inclusion. Importantly, when normalized by the radius of the inclusion $r_0$, the deformations $u/r_0$

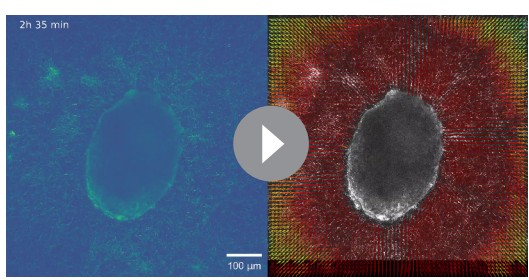

**Video 1.** Local matrix deformations around a Luminal B breast cancer spheroid. Left: Series of confocal reflection microscopy images of collagen fibers at the equatorial plane around a Luminal B breast cancer spheroid embedded in 1.2 mg/ml collagen gel. Time is indicated on the top left and measures time after initiation of collagen polymerization (by increasing the pH of the collagen solution to 10). The video starts once collagen fibers are becoming visible in the reflection channel (~30 min after initiation of polymerization). The default starting point of a traction force experiment is 60 min after polymerization started. Right: Measured deformation field surrounding the embedded spheroid as indicated by the color-coded arrows. The confocal reflection microscopy images are shown in gray-scale in the background. See *Figure 1—figure supplement 1* for a quantitative evaluation of this image series.

https://elifesciences.org/articles/51912#video1

collapse onto a single curve when plotted against the normalized distance $r/r_0$ (*Figure 2c*). This implies that the shape of the simulated deformation field only depends on the pressure but not on the size of the inclusion (i.e. on the spheroid radius $r_0$ at the time of seeding).

## Deformation fields in non-linear biopolymer networks

The collapse of the normalized deformation versus distance relationship furthermore implies that we can estimate the contractile pressure (contractile force per surface area) of a tumor spheroid of arbitrary size from a look-up table. To create this look-up table, we perform 150 simulations with pressures ranging from 0.1 Pa to 10,000 Pa. The simulated deformation fields are normalized by $r_0$, binned and interpolated to obtain smooth deformation curves (*Figure 3b*). For a low pressure of ~1 Pa, the deformation field as a function of radial distance from the spheroid center falls off with increasing distance according to a power law with an exponent (=slope in a double logarithmic plot) of $\alpha = -2$, as expected for a linear elastic material. With increasing pressure, however, the deformations near the spheroid surface fall off more slowly, with a slope approaching values around $\alpha = -0.2$ for high pressure values > 1000 Pa (*Figure 3—figure supplement 1*), indicating long-range force transmission due to a stiffening of the collagen fibers. This is in line with reported theoretical models (*Xu and Safran, 2015*; *Grimmer and Notbohm, 2018* and experimental findings (*Burkel and Notbohm, 2017*; *Han et al., 2018*).

To evaluate whether measured deformation fields match the predictions from simulation for different strains, we compare simulated and measured matrix deformations around a spheroid grown from 4000 primary triple-negative breast cancer cells over the course of 24 hr after embedding. To avoid tracking artifacts due to invading cells in the direct vicinity of the spheroid, we only use deformations that occur more than two radii away from the spheroid center. We find that triple-negative breast cancer cells deform the collagen matrix by ~200 μm (corresponding to a strain of over 50%; *Video 4*) near the spheroid surface after 24 hr of measurement time, resulting in a contractile pressure of 677 ± 68 Pa (median ± st.dev.) and a total contractility (pressure × surface area) of 344 ± 35 μN (median ± st.dev.; *Figure 3a,b*). At these high strains, collagen may experience plastic deformations and structural changes in addition to purely elastic deformations (*Kim et al., 2017*). Even though our material model only accounts for elastic deformations, we find excellent agreement between measured and simulated deformation fields (*Figure 3b*). Importantly, the simulations accurately capture the progressing flattening of the deformation field (deformation versus distance curves) due to strain stiffening of the matrix, which can exceed a 20-fold increase over the linear stiffness of collagen close to the surface of triple-negative breast cancer spheroids (*Video 5*).

## Far-field approximation for non-spherical objects

While spheroids are typically created from only one or two cell types, real tumors are more heterogeneous and contain tumor-generated matrix components as well as a mixture of epithelial and mesenchymal tumor cells that may split into subpopulations with different gene expression levels and gene mutations (*Shipitsin et al., 2007*). To investigate the interplay of different cell types and matrix components, we extract and isolate small samples from a patient-derived tumor and embed them in

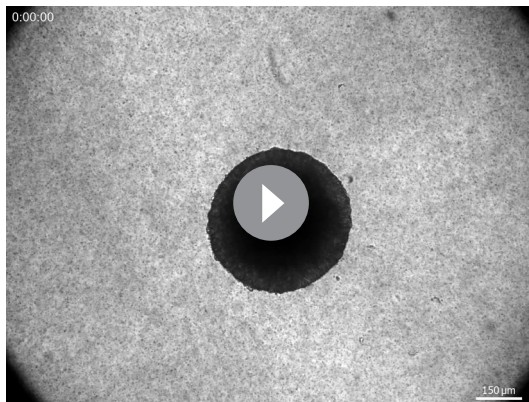

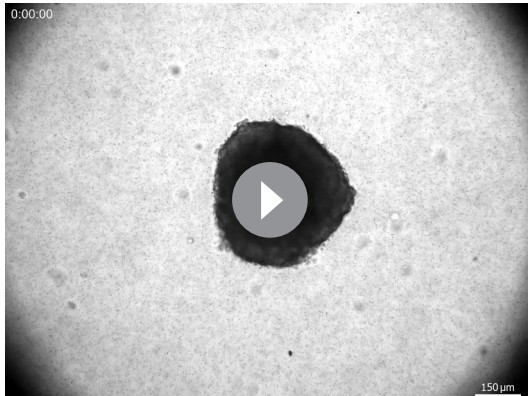

**Video 2.** Time-lapse brightfield images of an A172 glioblastoma spheroid generated from 15,000 cells embedded in a collagen gel over the time course of 12 h. Time is indicated in the upper-left corner (HH:MM:SS).

https://elifesciences.org/articles/51912#video2

**Video 3.** Time-lapse brightfield images of an U87 glioblastoma spheroid generated from 7,500 cells embedded in a collagen gel over the time course of 12 h. Time is indicated in the upper-left corner (HH:MM:SS).

https://elifesciences.org/articles/51912#video3

a collagen matrix. Due to the preparation process and their inherent heterogeneity, however, these tumoroids generally do not attain a high circularity in culture, but rather have a more elliptical, sometimes irregular shape.

To test whether our method is applicable to non-spherical contracting tissue samples, we apply the collagen contractility assay to tumoroids obtained from a Luminal B breast cancer patient (*Figure 4a–c*). We find that the tumoroids remain viable within the collagen gel for over 24 h and generate a median contractility of 24.5 µN (with a median effective radius of 149 µm; n = 14; *Video 6*). We find that for highly elongated tumoroids (*Figure 4a*), the material simulations overestimate the matrix deformations in the near-field ($r/r_0 \leq 4$), due to the oversimplified assumption of spherical geometry (*Figure 4d*). This may result in local deviations of the inferred pressure of up to 20% close to the spheroid (*Figure 4g*). Importantly, however, the simulated far-field deformations are still in good agreement with the measured deformations, irrespective of the pronounced eccentricity of the tumoroid (*Figure 4d,g*). For small tumoroids that only exert small absolute displace-

ments in the matrix (*Figure 4b*), we sporadically find local deviations in the inferred pressure of up to 20% at the outer rim of the field of view where the matrix deformations approach the resolution limit of the PIV algorithm (*Figure 4e,h*). Such outliers however do not significantly influence the inferred median pressure that takes into account the complete displacement field. For larger tumoroids with a more circular shape (*Figure 4c*), the local deviations from the inferred pressure are generally $\leq 5\%$.

As highly asymmetric tumoroids (and some spheroids) create asymmetric deformation fields in the surrounding matrix (and thereby an asymmetric stiffening of the matrix; *Video 7*), we further evaluate the directional contractile pressure by subdividing the deformation field around spheroids and tumoroids into narrow 5° angular segments. We find that the directional variability of the contractile pressure is equal to or smaller than the variability between individual tumoroids

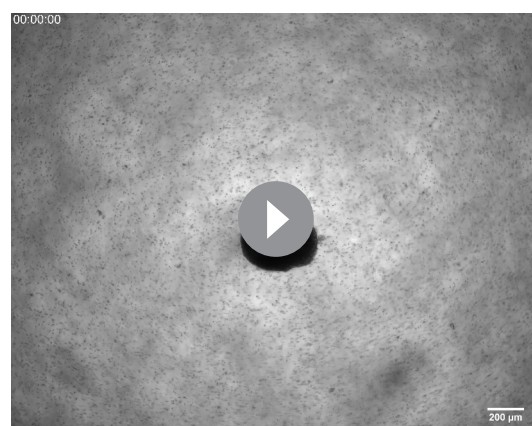

**Video 4.** Time-lapse brightfield images of a spheroid generated from 4000 primary triple-negative breast cancer cells embedded in a collagen gel over the time course of 24 h. Time is indicated in the upper-left corner (HH:MM:SS).

https://elifesciences.org/articles/51912#video4

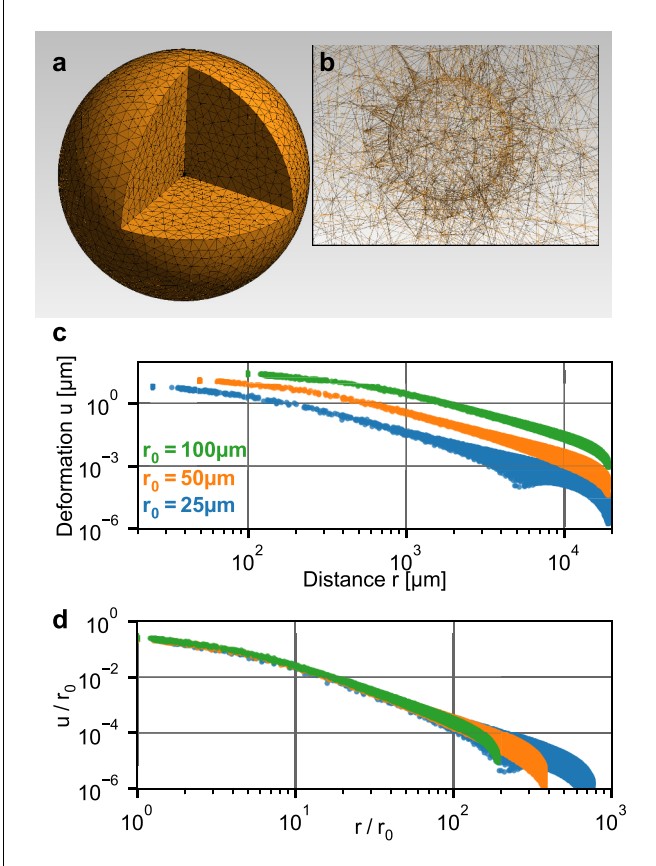

**Figure 2.** Simulation of a spherical inclusion in collagen. (**a**) Illustration of the tetrahedral mesh used for the material simulation. The spherical volume has a radius of 2 cm, with a spherical inclusion in the center. (**b**) Enlarged section of the tetrahedral mesh around the spherical inclusion with a radius of $r_0 = 100$ μm. c: Simulated absolute deformations $u(\vec{r})$ as a function of the distance $r = |\vec{r}|$ from the center of the volume, for an inward-directed pressure of 100 Pa acting on the surface of the inclusion. Different colors indicate different radii $r_0$ of the spherical inclusion. d: Same as in (**c**), but with deformations and distances normalized by $r_0$. For a given inbound pressure, all curves collapse onto a single relationship.

or spheroids (*Figure 4—figure supplement 1*, *Videos 8*, *9*, *10*, *11*). Thus, our method is applicable to non-spherical samples if we only consider the far-field deformations for force reconstruction.

## Mechanical feedback guides collective force generation

We next apply the collagen contractility assay to two glioblastoma cell lines, A172 (15,000 cells per spheroid) and U87 (7,500 cells per spheroid, to match the size of A172 spheroids; *Figure 5—figure supplements 1* and *2*) as little is known about the traction forces exerted by glioblastoma cells during the invasion of brain tissue. Although collagen is present in only small amounts in the normal human brain, glioblastoma cells readily bind to collagen (*Payne and Huang, 2013*). Recent studies have shown that fibrillar collagens are an integral part of the locally produced extracellular matrix in glioblastomas (*Huijbers et al., 2010*; *Pointer et al., 2017*). Furthermore, collagen is found in the basement membrane surrounding blood vessels, which are a major route of glioblastoma invasion (*Payne and Huang, 2013*; *Cuddapah et al., 2014*). Reconstituted collagen gels display a Young's modulus of $162 \pm 25$ Pa (*Steinwachs et al., 2016*) in the linear regime, closely emulating the soft environment of the brain tissue (100–1000 Pa *Levental et al., 2007*).

To investigate the role of collective effects in cellular force generation, we compare the contractility exerted by individual glioblastoma cells to the contractility of tumor spheroids generated from the respective cell lines. We apply single-cell 3D traction force microscopy as described in *Steinwachs et al. (2016)*. Specifically, we reconstruct the forces exerted by the cells on the collagen

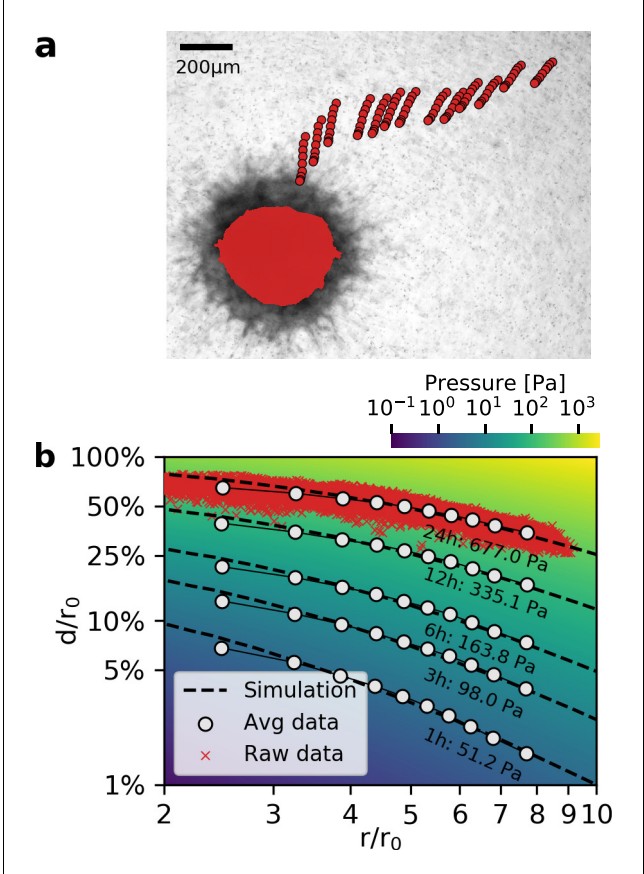

**Figure 3.** Deformation fields in non-linear biopolymer networks. (**a**) Brightfield image of a tumor spheroid grown from 4000 primary, triple-negative breast cancer cells, 24 hr after embedding in a 3D collagen gel together with fiducial markers. The initial shape of the spheroid at the beginning of the experiment is indicated by the red shading. Red circles show the trajectory of exemplary fiducial markers over the course of 24 hr measurement time to illustrate the material strain arising within the matrix due to the contractile force of the spheroid. b: Normalized deformations as a function of the normalized distance for material simulations of varying pressure (color coding). Each red marker corresponds to the normalized deformation within an individual image tile analyzed with particle image velocimetry, after 24 hr measurement time. White circles indicate averaged normalized deformations for different time points during the measurement (times and inferred pressure values are noted below each curve). Dashed black lines indicate the corresponding best-fit simulated deformation field.

The online version of this article includes the following figure supplement(s) for figure 3:

**Figure supplement 1.** Power-law scaling of deformation fields.

gel from the surrounding deformation field (*Figure 5a,b*). By summing up all force components that point toward the force epicenter, we obtain the total contractility. We find that individual A172 cells are nearly 2-fold stronger compared to U87 cells (91 nN vs. 51 nN; *Figure 5c*).

In the case of spheroids generated from A172 and U87 cells, we find that the collective contractility observed at an early time point (30 min after the beginning of the measurements) closely reflects the differences seen at an individual cell level: A172 spheroids are nearly 2-fold stronger compared to equally sized U87 spheroids (21 µN vs. 11 µN; *Figure 5d*). During these initial time steps, the induced strains on the collagen matrix are still small, and hence there is no global stiffening of the material which could feed back to cell behavior. By contrast, after 12 h, A172 spheroids and U87 spheroids generate comparable collective contractilities of 140 µN and 149 µN, respectively (*Figure 5e*). While U87 spheroids keep increasing their contractility over the complete 12 h observation period, A172 spheroids show a 2 h-resting period of after a fast initial increase in contractility (*Figure 5f*, *Figure 5—figure supplement 2*). Such a collective resting period requires a synchronized change in cellular force generation across the whole cell population. A likely mediator for this cell-

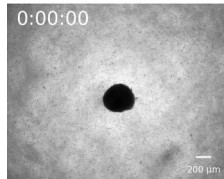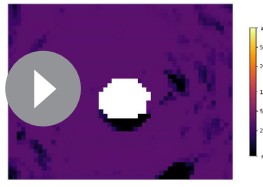

**Video 5.** Local matrix stiffness in the vicinity of a triple-negative breast cancer spheroid. Left: Time-lapse brightfield images of a spheroid generated from 4000 primary triple-negative breast cancer cells embedded in a collagen gel over the time course of 24 h. Time is indicated in the upper-left corner (HH:MM:SS). Right: Local stiffness map of the collagen matrix surrounding the spheroid. Stiffness is displayed on a logarithmic scale and calculated in radial direction relative to the spheroid center. At zero strain, the 1.2 mg/ml collagen gel has a stiffness of 316 Pa. After 24 h measurement time, the maximum local stiffness is 7585 Pa.

https://elifesciences.org/articles/51912#video5

cell coupling is the collagen matrix: as the cells pull on the matrix, collagen exhibits strain stiffening. This change in material stiffness then provides a mechanical feedback to the cells and may thus alter cell behavior at the population level (*Morley et al., 2019*). This example illustrates that collective contractility is not necessarily related to the respective traction forces of individual cells, especially over longer time scales.

## Collective twitching in tumor spheroids

In a previous study (*Steinwachs et al., 2016*), we have shown that the contractility of individual breast carcinoma cells varies significantly over time, with alternating phases of low and high contractility that last for 50 min on average and that correlate with the migration process of these cells. By contrast, the data reported above demonstrate that spheroids generated from primary triple-negative breast cancer cells, U87 and A172 glioblastoma cells, and Luminal B breast tumoroids all increase their contractility monotonically over time. However, spheroids made from primary Luminal B breast cancer cells show a different behavior: after 2 h of measurement time, these spheroids begin to show repeated twitches (*Figure 6a*, *Video 12*) during which the spheroid relaxes the matrix and subsequently contracts again. These contractile twitches are synchronized across the whole spheroid and thus lead to isotropic, radially symmetric inward-outward movements of the surrounding matrix (*Figure 6b,c*). An individual twitch is completed after 20 min (*Figure 6a* inset), indicating a fast-moving signal as a mediator of the effect. The amplitude of individual twitches is in between 2–20 µN around a total contractility of 200–400 µN, demonstrating the ability of our method to measure dynamic force fluctuations with relative changes as small as 1%.

## Discussion

In this study, we develop, test and apply a contractility assay for quantifying the collective force generation process in tumor spheroids containing hundreds or thousands of cells. Because the assay takes the pronounced strain stiffening of a collagen matrix into consideration, simulated and measured deformation fields in the collagen matrix surrounding a spheroid show good agreement even for large contractile forces with strains of >50% at the spheroid surface. While our method relies on the assumption of a spherical sample geometry, it remains accurate in the far field (>3–4 sample radii) for elliptical or irregularly shaped tumoroids.

For A172 and U87 glioblastoma cells, we find that the collective forces are proportional to the contractility of individual cells during the initial contraction phase ($\leq$ 1 h), but not on longer time scales. In particular, the large strains induced by the spheroids significantly alter the mechanical environment of the invading cells due to strain stiffening and fiber alignment, and thus affect cellular force generation at a collective level and potentially induce enhanced invasion into the surrounding tissue.

Finally, we report collective twitching of spheroids generated from primary Luminal B cells. While the origin of this effect remains unknown, it demonstrates that these cells are able to synchronize their force generation across an entire spheroid containing several thousands of cells. We note that twitching starts only after the spheroid has already generated appreciable matrix deformations approximately 12 h after the beginning of the measurements, corresponding to a contractility of 200 µN or larger. As cell-matrix interactions and the process of tissue remodelling are increasingly recognized as therapeutic targets (*Cox and Erler, 2011*), our method provides a reliable and simple

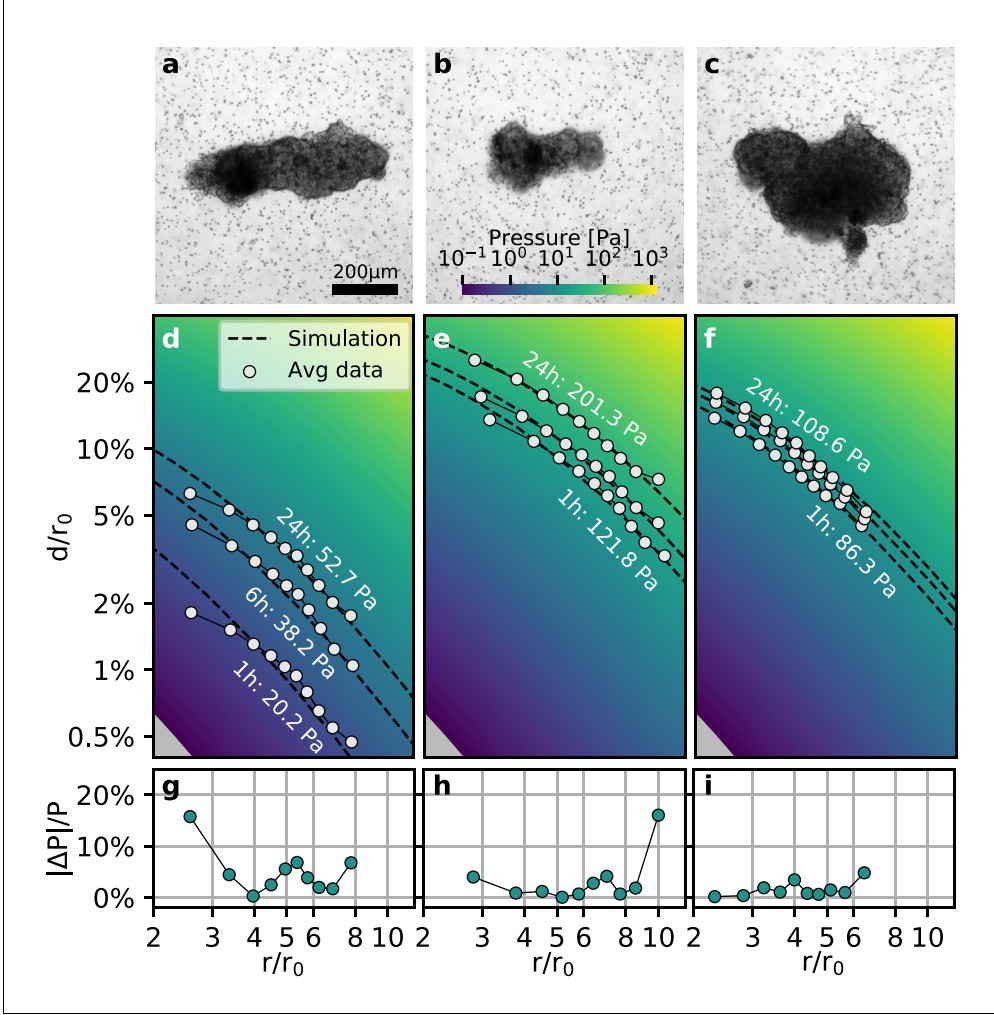

**Figure 4.** Collective contractility of Luminal B breast tumoroids. (**a-c**) Brightfield images of three exemplary tumoroids embedded in a 3D collagen matrix, together with fiducial markers. (**d-f**) Normalized averaged measured matrix deformations (white circles) of the corresponding tumoroids (**a–c**) for three time points (1 h, 6 h, and 24 h) after the beginning of the experiment. The dashed lines indicate the corresponding best-fit deformation field from the material simulations. The color-coded background indicates simulated deformations for a range of pressures. g-i: Local relative deviation of the inferred pressure from the best-fit pressure value. Larger values indicate that the measured displacement field deviates stronger from the simulated displacement field.

The online version of this article includes the following figure supplement(s) for figure 4:

**Figure supplement 1.** Angle-dependent variation of reconstructed pressure values.

in vitro assay to quantify the mechanics behind collective effects in cancer invasion that cannot be measured on a single-cell level.

## Materials and methods

### Primary breast tumoroid and primary cell line isolation

Human tissue collection was approved by the Ethics Committee of the Friedrich-Alexander University Erlangen-Nürnberg, Germany (#99_15Bc) in accordance with the World Medical Association Declaration of Helsinki. Informed consent was obtained from all patients.

The Luminal B tumor was obtained from a patient with Luminal B lymph node positive breast cancer (20% Ki67 positive, hormone receptor positive, but Her2 receptor negative) and no prior chemotherapy. The Triple Negative tumor was obtained from a breast cancer patient (70% Ki67 positive,

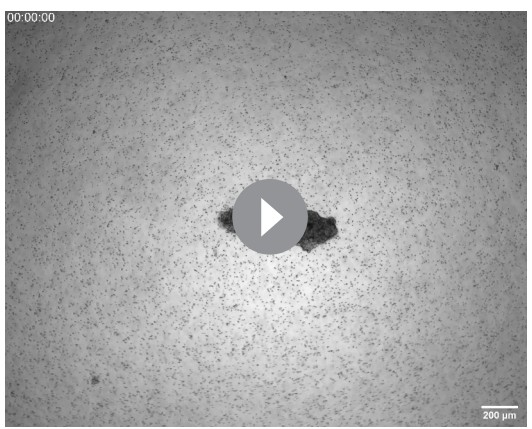

**Video 6.** Time-lapse brightfield images of a Luminal B tumoroid embedded in a collagen gel over the time course of 24 h. Time is indicated in the upper-left corner (HH:MM:SS).

https://elifesciences.org/articles/51912#video6

estrogen receptor, progesterone receptor and Her2 receptor negative; and no prior chemotherapy; *Weigand et al., 2016*).

Isolation of primary breast tumoroids and primary cell lines are performed as described in *Weigand et al. (2016)*. In brief, all breast tumors are examined by a pathologist, the tumor cores identified, dissected, washed 4x with 1x PBS and then incubated with 1x PBS, penicillin and streptomycin for 1 hr at room temperature. Following tissue mincing and an overnight digestion with collagenase/hyaluronidase (Stem Cell Technologies) in basal culture media, the cell lysate is diluted 1:1 with 1x PBS and centrifuged at 88 g for 30 sec at room temperature. The pellet fraction contains tumoroids, which are either purified further into a single fraction or further fractionated into either epithelial or cancer mesenchymal cells (*Weigand et al., 2016*). All primary cell lines used in this present study are breast cancer mesenchymal cells. To obtain a purified fraction of tumoroids, the initial pellet is resuspended into 5 ml of 1x PBS then processed 5x with a 1.00 × 60 mm sterile needle, diluted 10x with 1x PBS and filtered through 100 µm nylon filters (10 ml per filter; Falcon) and then washed 2x to separate fibrotic tissue. The final tumoroid size ranges from 200 to 600 µm in diameter.

## Primary breast tumoroids and primary breast cell line culture

Directly following isolation, tumoroids are cultured for 4 days in Epicult basal media and Supplement C (Stem Cell Technologies; Epicult-C human media kit), and L-Glutamine on top of 2% soft agarose beds in 2 cm$^2$ culture dishes. This incubation period promotes tumoroid recovery following the isolation procedure from primary tissue. Established primary breast cancer mesenchymal cells are isolated from two Luminal B (LUB1, LUB25) and a Triple negative (TRIDUC1 *Weigand et al., 2016*) breast cancer and cultured in Epicult basal media with Supplement C (Stem Cell

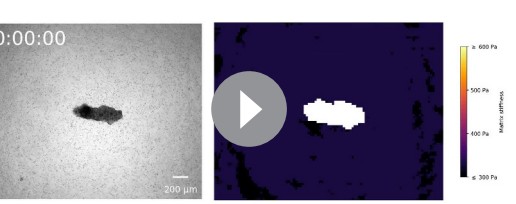

**Video 7.** Local matrix stiffness in the vicinity of a triple-negative breast cancer spheroid. Left: Time-lapse brightfield images of a Luminal B tumoroid embedded in a collagen gel over the time course of 24 h. Time is indicated in the upper-left corner (HH:MM:SS). Right: Local stiffness map of the collagen matrix surrounding the tumoroid. Stiffness is displayed on a linear scale and calculated in radial direction relative to the spheroid center. At zero strain, the 1.2 mg/ml collagen gel has a stiffness of 316 Pa. After 24 h measurement time, the maximum local stiffness is 516 Pa.

https://elifesciences.org/articles/51912#video7

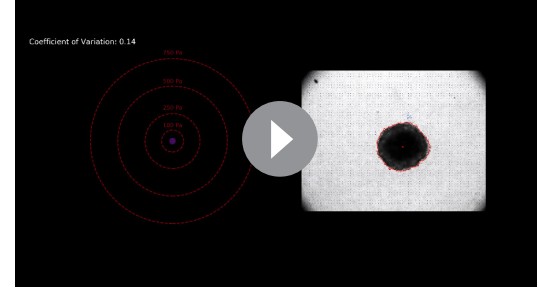

**Video 8.** Angular contractile pressure of a glioblastoma spheroid. Left: Angular dependence of contractile pressure of the glioblastoma spheroid shown in *Figure 1 d–g*. Each point represents the reconstructed contractile pressure from a 5°-segment of the deformation field surrounding the spheroid. The coefficient of variation is defined as mean/st.dev. and denotes the variation of the reconstructed contractile pressure between different directions. Right: Time-lapse images of the equatorial plane of the spheroid. Matrix deformations are shown as arrows, the initial spheroid outline is indicated in red. Images are synchronized to the pressure values shown on the left.

https://elifesciences.org/articles/51912#video8

Technologies, Epicult-C human media kit), L-Glutamine, 10% FCS and then initiated into spheroid formation at an early cell passage number of 3–4.

## Glioblastoma cell line culture

A172 and U-87 MG (referred to as U87 in the main text) glioblastoma cell lines are cultured at 37°C, 95% humidity and 5% $CO_2$ in DMEM (high glucose, pyruvate) with 10% (volume/volume) fetal bovine serum, and 100 Units/ml penicillin/streptomycin (all Thermo Fisher Scientific). Cell lines are short tandem repeat (STR) profiled to confirm identity (CellBank Australia) and are confirmed negative for mycoplasma contamination with Venor GeM Classic detection kit (Minerva biolabs).

## Spheroid culture

Glioblastoma spheroids are created from low-adherent, concave-bottomed surfaces in 96-well dishes (Friedrich et al., 2009). 50 µl of a heated 1.5% (weight/volume) agarose (Thermo Fisher Scientific)/DMEM gel solution is pipetted into the wells of a 96-well dish. Following a 10–15 min interval, the solution cools and forms a non-adherent, concave surface. Subsequently, cells are detached from their tissue culture flasks with 0.05% trypsin solution, counted (15,000 cells per dish for A172 cells and 7,500 cells per dish for U87 cells) and pipetted into wells containing 100 µl cell culture medium. The agarose surface promotes formation of a single spheroid per well. Spheroids take 3 days to fully form while being incubated at standard TC conditions.

Primary breast cancer spheroids are created from cell-repellent, U-bottom, 96-well dishes (Greiner). Cells are detached from their tissue culture flasks with 0.05% trypsin solution, counted (4,000 cells per dish for LUB1, LUB25 and TRIDUC1 cells) and pipetted into wells containing 100 µl cell culture medium. Spheroids take 2 days to fully form while being incubated at standard TC conditions.

## Collagen synthesis

Collagen gels are synthesized as described in Steinwachs et al. (2016) and consist of a 1:1 mixture of rat tail collagen (Collagen R, 2 mg/ml, Matrix Bioscience, Berlin, Germany) and bovine skin collagen (Collagen G, 4 mg/ml, Matrix Bioscience), plus 10% (vol/vol) $NaHCO_3$ (23 mg/ml) and 10% (vol/vol) 10 × DMEM (Gibco). The pH of the solution is adjusted to 10 with 1 M NaOH. For a collagen concentration of 1.2 mg/ml, the solution is diluted with a mixture of 1 vol part $NaHCO_3$, 1 part 10 × DMEM and 8 parts $H_2O$, at a ratio of 1:1.

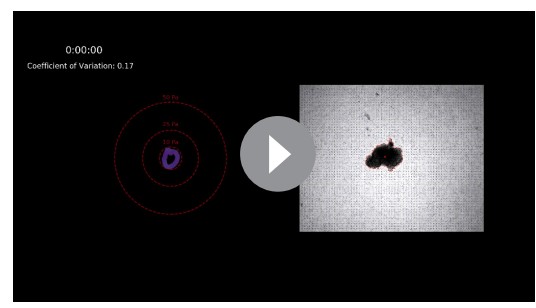

**Video 9.** Angular contractile pressure of a Luminal B breast cancer spheroid. Left: Angular dependence of contractile pressure of an exemplary Luminal B breast cancer spheroid. Each point represents the reconstructed contractile pressure from a 5°-segment of the deformation field surrounding the spheroid. The coefficient of variation is defined as mean/st.dev. and denotes the variation of the reconstructed contractile pressure between different directions. Right: Time-lapse images of the equatorial plane of the spheroid. Matrix deformations are shown as arrows, the initial spheroid outline is indicated in red. Images are synchronized to the pressure values shown on the left.
https://elifesciences.org/articles/51912#video9

**Video 10.** Angular contractile pressure of a Luminal B tumoroid. Left: Angular dependence of contractile pressure of an exemplary Luminal B tumoroid. Each point represents the reconstructed contractile pressure from a 5°-segment of the deformation field surrounding the tumoroid. The coefficient of variation is defined as mean/st.dev. and denotes the variation of the reconstructed contractile pressure between different directions. Right: Time-lapse images of the equatorial plane of the tumoroid. Matrix deformations are shown as arrows, the initial spheroid outline is indicated in red. Images are synchronized to the pressure values shown on the left.
https://elifesciences.org/articles/51912#video10

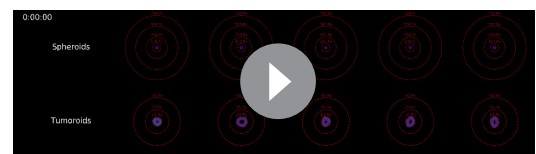

**Video 11.** Angular dependence of contractile pressure of Luminal B breast cancer spheroids (top row) and tumoroids (bottom row) over the course of 22 h. Each point represents the reconstructed contractile pressure from a 5°-segment of the deformation field surrounding the spheroid/tumoroid.

https://elifesciences.org/articles/51912#video11

## Glioblastoma spheroid embedding

FluoSphere polystyrene beads (1 µm diameter, Thermo Fisher Scientific) are carefully suspended, without forming bubbles, in 1.2 mg/ml collagen solution at a concentration of $2 \cdot 10^8$ beads/ml. 1.5 ml of this mixture is poured into a 35 mm plastic culture dish and is allowed to settle for 2.5 min at room temperature, during which time the spheroids are prepared for embedding. The 2.5 min waiting time is too short for a full polymerization of the collagen solution but is sufficient to ensure that spheroids do not sink to the base of the dish.

After the preparation of the bottom collagen layer, 4 to 5 individual spheroids are removed from their culture plate wells and carefully transferred into a 15 ml centrifuge tube using a P1000 pipette. Once the spheroids have settled to the base of the tube, excess media is aspirated away and spheroids are gently resuspended in 500 µl of the 1.2 mg/ml collagen/bead mixture. The mixture, complete with suspended spheroids, is then transferred from the tube into the 35 mm dish using a P1000 pipette. By pipetting the collagen into the dish drop-by-drop, the positioning of the

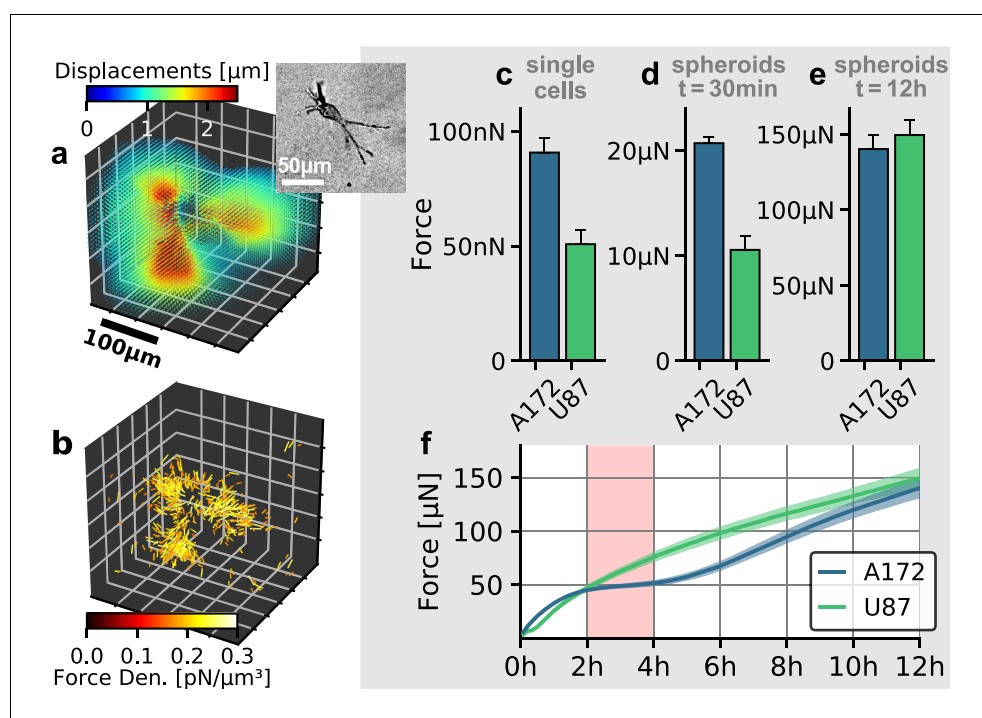

**Figure 5.** Individual and collective contractility of glioblastoma cells. (a) Matrix deformations exerted by an exemplary A172 cell (inset) embedded in a 3D collagen gel. (b) Reconstructed force density field surrounding the A172 cell shown in (a). (c) Median cell contractility as measured by single-cell 3D traction force microscopy (A172: n = 90; U87: n = 86). (d) Mean collective cell contractility of tumor spheroids after 30 min measurement time (A172: n = 17; U87: n = 13). e: Mean collective cell contractility of tumor spheroids after 12 h measurement time (A172: n = 17; U87: n = 13). f: Time course of the mean contractility and corresponding standard error (shaded) for A172 (blue) and U87 (green) spheroids. The 2 h-resting period of the A172 spheroids is marked in red. Error bars denote 1 standard error.

The online version of this article includes the following figure supplement(s) for figure 5:

**Figure supplement 1.** Glioblastoma spheroid shape and size.
**Figure supplement 2.** Glioblastoma spheroid contractility.

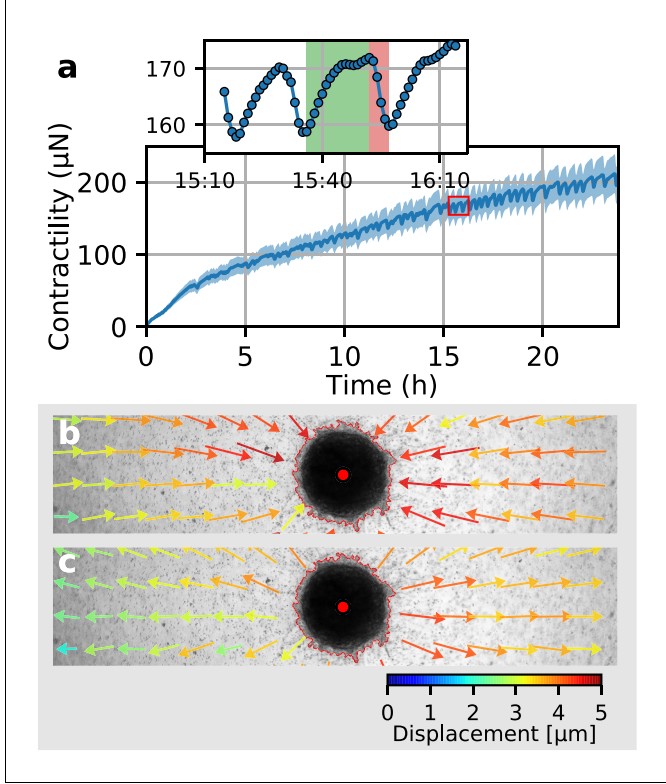

**Figure 6.** Collective pulsing in Luminal B breast cancer spheroids. (**a**) Median contractility (blue line) and the corresponding standard deviation (blue shading) of an exemplary spheroid grown from 4000 primary Luminal B breast cancer cells. The red box marks the contractility values displayed in the inset, illustrating a single twitch starting at 15:35 hr after the beginning of the measurement. (**b**) Changes in matrix deformations during the contraction phase of a single twitch that lasts for 15 min and is marked in green in (**a**). Inward-directed arrows indicate increasing contractility. The spheroid outline and its centroid are marked in red. (**c**) Changes in matrix deformations during the relaxation phase of a single twitch that lasts for 5 min and is marked in red in (**a**). Outward-directed arrows indicate decreasing contractility.

spheroids within the gel can be controlled. Spheroids are kept separate from each other and away from culture dish margins or air bubbles. After spheroid seeding, the gel is incubated at 37 °C and 5% $CO_2$ for 1 h to fully polymerize. 1.5 ml of pre-warmed cell media is added to the dish, and imaging is started.

## Primary breast cancer spheroid and tumoroid embedding

Silica beads (4 μm diameter, Kisker Biotech) are carefully suspended, without forming bubbles, in 1.2 mg/ml collagen solution at a concentration of $2 \cdot 10^8$ beads/ml. 1300 μl of this mixture is poured into one well of a 6-well plate and is allowed to settle for 10 min in the incubator, during which time the spheroids are prepared for embedding. The 10 min waiting time results in a partially polymerized collagen surface and thus ensures that spheroids/tumoroids do not sink to the base of the dish.

After the preparation of the bottom collagen layer, up to 10 individual spheroids/tumoroids are removed from their culture plate wells and carefully transferred into a 15 ml centrifuge tube using a P200 pipette. Once the spheroids/tumoroids have settled to the base of the tube, excess media is aspirated away and spheroids/tumoroids are resuspended in 1300 μl of the 1.2 mg/ml collagen/bead mixture. The mixture, complete with suspended spheroids/tumoroids, is then transferred from the tube into the well using a P1000 pipette. After spheroid seeding, the gel is incubated at 37 °C and 5% $CO_2$ for 1 h to fully polymerize. 2 ml of pre-warmed cell media is added to the well, and imaging is started.

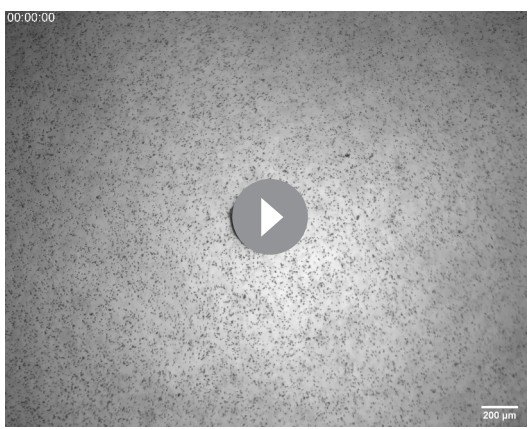

**Video 12.** Time-lapse brightfield images of a spheroid generated from 4000 primary Luminal B breast cancer cells embedded in a collagen gel over the time course of 24 h. Time is indicated in the upper-left corner (HH:MM:SS).

https://elifesciences.org/articles/51912#video12

## Time-lapse imaging

The equatorial plane of the embedded glioblastoma spheroids is imaged in brightfield mode with a 5x magnification 0.1 NA objective and a CCD camera (corresponding to a pixel size of 1.29 μm) for at least 12 h, with a time interval of 5 min between consecutive images. Samples are kept in a stage-mounted incubation chamber (37 °C, 5% $CO_2$) during time-lapse imaging. Typically, 3–7 spheroids are imaged in parallel (contained in one dish). In total, we imaged 17 A172 spheroids with 15,000 cells, 14 U87 spheroids with 15,000 cells, and 13 U87 spheroids with 7,500 cells (at least three independent experiments per condition; *Figure 5—figure supplement 2*).

Primary breast cancer spheroids and tumoroids are imaged using a 4x magnification 0.13 NA objective and a CCD camera (corresponding to a pixel size of 1.02 μm) for 24 h, with a time interval of 10 min between consecutive images (except for the measurement of the pulsing Luminal B spheroid shown in *Figure 6*, which was recorded at a time interval of 1 min). Samples are imaged at 37 °C and 5% $CO_2$ using a microscope placed inside an incubator. We performed measurements on three primary triple-negative breast cancer spheroids on the same day. From those three measurements, one exemplary data set is shown in *Figure 3*. We performed measurements on five primary Luminal B breast cancer spheroids (LUB25) on the same day. These data sets are shown in the Supplementary Information. We performed measurements on five primary Luminal B breast cancer spheroids (LUB1) on the same day. From those five measurements, one exemplary data set is shown in *Figure 6*. We performed measurements on 14 tumoroids from the same patient on the same day. From those 14 measurements, exemplary data sets from three tumoroids are shown in *Figure 4*.

## Material simulations

We use the semi-affine material model described in *Steinwachs et al. (2016)* to simulate the non-linear behavior of collagen. In particular, collagen gels exhibit three different mechanical regimes, depending on the applied strain. Individual fibers buckle easily under compression (with exponentially suppressed stiffness) and only attain a constant stiffness for small strains, while they exponentially stiffen under large strains:

$$\kappa(\varepsilon) = \kappa_0 \cdot \begin{cases} e^{\varepsilon/d_0} & \text{for } \varepsilon < 0 & \text{buckling} \\ 1 & \text{for } 0 < \varepsilon < \varepsilon_s & \text{linear regime} \\ e^{(\varepsilon - \varepsilon_s)/d_s} & \text{for } \varepsilon_s < \varepsilon & \text{strain stiffening} \end{cases} \tag{1}$$

where $\kappa_0$ denotes the linear stiffness, $d_0$ and $d_s$ describe the rate of stiffness variation during buckling and stiffening, respectively, and $\varepsilon_s$ denotes the onset of strain stiffening.

These four parameters can be characterized by shear rheometry and by measuring the vertical contraction of a collagen gel under uniaxial stretch. More specifically, the experimentally obtained stress-strain curve from shear rheometry and the contraction-stretch curve from the uniaxial stretch-experiment are fitted to the semi-affine material model described above. The open-source software saenopy provides ready-to-use fitting routines for this purpose, see https://saenopy.readthedocs.io/.

In this study, we use the material parameters determined in *Steinwachs et al. (2016)*, for a 1.2 mg/ml collagen solution based on a 1:1 mixture of rat tail collagen and bovine skin collagen:

$$\kappa_0 = 1645 \, \text{Pa}, \varepsilon_s = 0.0075, d_s = 0.033, d_0 = 0.0008 \tag{2}$$

Note that for different collagen concentrations, only the linear stiffness needs to be adjusted (0.6 mg/ml: $\kappa_0 = 447\,\mathrm{Pa}$, 2.4 mg/ml: $\kappa_0 = 5208\,\mathrm{Pa}$), whereas $d_0$, $d_s$, $\varepsilon_s$ remain constant.

Deformations in the collagen matrix in response to inward-directed tractions at the spheroid surface are computed using a finite element approach (*Steinwachs et al., 2016*). In brief, the material volume is divided into finite tetrahedral elements, each of which is assumed to contain a number of isotropically oriented fibers. When such a tetrahedron is deformed, the internal stress is first calculated by taking into account the different deformations of the contained fibers, and subsequently averaged over the faces of the tetrahedron and thus propagated to neighboring elements.

Here, we simulate a spherical bulk of material (with an outer radius of 2 cm) with a small spherical inclusion in its center (with a radius of 100 µm). The finite element mesh for this geometry is created using the open-source software Gmsh (*Geuzaine and Remacle, 2009*). To emulate the contractile behavior of a spheroid, we assume a constant inbound pressure on the surface of the spherical inclusion and further assume zero deformations on the outer boundary of the bulk. Given these boundary conditions, we use the Python-port of the open-source Semi-Affine Elastic Network Optimizer (*Steinwachs et al., 2016*) (saenopy) to obtain the corresponding deformation field.

## Particle image velocimetry

Given a series of images through the equatorial plane of the spheroid, we apply the open-source PIV software (OpenPIV *Taylor et al., 2010*) to each pair of subsequent images. The software subdivides the image recorded at time $t$ into $N$ quadratic tiles (using a tile-size of $40 \times 40$ pixels for the glioblastoma spheroids and $50 \times 50$ pixels for primary breast cancer spheroids and tumoroids) at positions $\vec{x}^{(i)}$ with $i = 1, 2, ..., N$ and performs a cross-correlation-based template-matching to determine the most likely offset $\Delta \vec{u}_t^{(i)}$ of all tiles with respect to the previous image. These offsets represent the deformation of the material within the time interval between two subsequent images. To account for a drift of the microscope stage between two images, we subtract the mean value

$$\vec{\mu}_t = \frac{1}{N} \sum_{i=1}^{N} \Delta \vec{u}_t^{(i)} \tag{3}$$

from all offsets for a given time step. To obtain the accumulated deformation $\vec{u}_t^{(i)}$ at position $\vec{x}^{(i)}$ and time step $t$, we sum up the pair-wise deformation fields of all time steps $t' \leq t$:

$$\vec{u}_t^{(i)} = \sum_{t'=1}^{t'=t} \Delta \vec{u}_{t'}^{(i)} \text{ for } i = 1, 2, ..., N \tag{4}$$

Additionally, we determine the spheroid's centroid $\vec{x}_t^{\mathrm{sph}}$ for all time steps and its initial radius $r_0$ by image segmentation (using *Otsu, 1979* method). As we are only interested in the radially aligned deformations towards the contracting spheroid, we compute the absolute deformations $u_t^{(i)}$ by projecting the accumulated vectorial deformations $\vec{u}_t$ in the direction towards the spheroid center, using the relative coordinates $\vec{d}_t^{(i)} = \vec{x}^{(i)} - \vec{x}_t^{\mathrm{sph}}$:

$$u_t^{(i)} = -\left( \vec{u}_t^{(i)} \cdot \frac{\vec{d}_t^{(i)}}{|\vec{d}_t^{(i)}|} \right), \tag{5}$$

where ( . ) denotes the dot product. While we place individual spheroids as far apart as possible within the collagen matrix, other spheroids outside the field-of-view may still affect the measured deformation field, especially in the case of small deformations within the field-of-view. To minimize this systematic bias, we only consider projected deformations that point within a ±20° range towards the spheroid center, by imposing the following condition:

$$\left| \frac{\vec{u}_t^{(i)}}{|\vec{u}_t^{(i)}|} \cdot \frac{\vec{d}_t^{(i)}}{|\vec{d}_t^{(i)}|} \right| > \cos(20°) \tag{6}$$

Finally, we compute the normalized deformations $u_t^{(i)}/r_0$ and distances $d_t^{(i)}/r_0$ so that we can directly compare to experimentally measured and normalized deformation fields.

### Geometrical scaling in nonlinear elastic materials

The scale-invariance of the deformation field of a contracting spherical inclusion within a large body of non-linear elastic material (shown by simulation in *Figure 2* in the main text) can be derived analytically. The following derivation is discussed in more detail in *Steinwachs (2015)*.

Given a displacement field $\vec{U}(\vec{r})$ of an equilibrium configuration (e.g. induced by a spheroid with a radius of 100 µm and an inbound pressure of 50 Pa), we define the re-scaled displacement field (e.g. induced by a spheroid with a radius of 200 µm and the same pressure of 50 Pa) as

$$\vec{U}^{*}(\vec{r}) = a \cdot \vec{U}(\vec{r}/a) \tag{7}$$

with $a$ being the scaling factor ($a = 2$ for the exemplary spheroid sizes noted above). To check whether the equilibrium state remains unaltered by re-scaling the displacement field in this way, we need to show that the deformation gradient $\underline{F}(\vec{r})$ and thus the strain energy density $W(\vec{r})$ as well as the nominal stress $\underline{N}(\vec{r})$ are not altered by the transformation (at the corresponding points $\vec{r} \to \vec{r}/a$):

$$\underline{F}^{*}(\vec{r}) = \frac{\partial \vec{U}^{*}(\vec{r})}{\partial \vec{r}} + \underline{I} = a \cdot \frac{\partial \vec{U}(\vec{r}/a)}{\partial \vec{r}} + \underline{I} = \underline{F}(\vec{r}/a) \tag{8}$$

$$\longrightarrow W^{*}(\vec{r}) = W(\vec{r}/a) \tag{9}$$

$$\longrightarrow \underline{N}^{*}(\vec{r}) = \underline{N}^{*}(\vec{r}/a) \longrightarrow \mathrm{div}(\underline{N}^{*}(\vec{r})) = \frac{1}{a} \cdot \mathrm{div}(\underline{N}^{*}(\vec{r}/a)), \tag{10}$$

where $\underline{I}$ denotes the unit tensor. Consequently, the equilibrium equation is fulfilled if the body force $\vec{b}(\vec{r})$ is divided by the scaling factor $a$:

$$\rho_0 \vec{b}^{*}(\vec{r}) + \mathrm{div}(\underline{N}^{*}(\vec{r})) = 0 = \frac{1}{a} \cdot \left( \rho_0 \vec{b}^{*}(\vec{r}/a) + \mathrm{div}(\underline{N}^{*}(\vec{r}/a)) \right) \tag{11}$$

We next consider an infinite continuous body with a spherical hole of radius $R$ at the origin. As a boundary condition, we assume that the spherical inclusion has decreased its radius by $\Delta R$, and we denote the displacement field $\vec{U}(\vec{r})$ as the equilibrium solution. The total strain energy needed for the inclusion to contract (or dilate) can be determined by integrating the strain energy density:

$$E(\Delta R) = \int_{R}^{\infty} W(\vec{r}) d^3 \vec{r} \tag{12}$$

If we now assume that a spherical inclusion with a radius $a \cdot R$ contracts by $a \cdot \Delta R$, we can use the scaling laws noted above to relate the strain energy to that of the un-scaled contracting inclusion:

$$E^{*} = \int_{a \cdot R}^{\infty} W(\vec{r}/a) d^3 \vec{r} = a^3 \cdot \int_{R}^{\infty} W(\vec{r}) d^3 \vec{r} = a^3 \cdot E \tag{13}$$

In equilibrium, the strain energy of the contracted inclusion depends only on $R$, $\Delta R$, and the scaling factor $a$:

$$E(a \cdot R, a \cdot \Delta R) = a^3 \cdot E(R, \Delta R) \tag{14}$$

Finally, we show that the normal surface pressure $P$ induced by the contraction of the spherical inclusion only depends on the relative contraction $\Delta R/R$, but not on the scaling factor $a$:

$$\frac{\partial E(R, \Delta R)}{\partial \Delta R} \cdot \frac{1}{4\pi R^2} = \frac{\frac{\partial E(a \cdot R, a \cdot \Delta R)}{a^3}}{\frac{\partial (a \cdot \Delta R)}{a}} \cdot \frac{1}{4\pi R^2} = \frac{\partial E(a \cdot R, a \cdot \Delta R)}{\partial (a \cdot \Delta R)} \cdot \frac{1}{4\pi a^2 R^2} = P(\Delta R/R) \tag{15}$$

Given a fixed surface pressure, a simulated displacement field $\vec{U}(\vec{r})$ of a spherical inclusion with radius $R$ is thus directly related to the deformation field $\vec{U}^{*}(\vec{r})$ of a spherical inclusion with radius $R^{*}$ by proper re-scaling with the factor $a = R^{*}/R$.

## Force reconstruction

To assign a contractility value to a measured deformation field, we first conduct 150 material simulations assuming an inbound pressure on the surface of the spherical inclusion ranging from 0.1 Pa to 10,000 Pa (logarithmically spaced), and interpolate between the resulting deformations fields to create a look-up function that translates any deformation/distance-tuple to a best-fit pressure value. For each measured deformation vector (projected towards the center of the spheroid), we then assign the best-fit pressure value. Finally, we take the median of all assigned pressure values at a given time step to obtain a single pressure value for an individual spheroid.

To obtain the contractility $F$ of an individual spheroid from the contractile pressure $P$, we need to account for the spheroid surface area $A$. As we only have images of the equatorial plane of the spheroid, we approximate the surface area by computing an effective radius $r$ from the top-view projected spheroid area $A_{\mathrm{proj}}$:

$$A = 4\pi r^2 \,\mathrm{with}\, r = \sqrt{\frac{A_{\mathrm{proj}}}{\pi}} \tag{16}$$

The projected spheroid area is determined at the beginning of the experiment by image segmentation (using *Otsu, 1979* method).

We provide the Python package jointforces that implements this force reconstruction method. The package further provides pre-computed look-up functions for different collagen gel concentrations (0.6 mg/ml, 1.2 mg/ml, 2.4 mg/ml) as well as for fibrin gel (4.0 mg/ml) and Matrigel (10 mg/ml). See https://github.com/christophmark/jointforces.

## Local matrix stiffness

To determine the local changes in matrix stiffness (see *Videos 5* and *7*), we first determine the radial strain $\varepsilon$ of the material from the deformation field:

$$\varepsilon(\vec{r}) = \frac{u(\vec{r}) - u(\vec{r} + \delta \cdot \vec{e}_r(\vec{r}))}{\delta}, \tag{17}$$

where $u(\vec{r})$ denotes the matrix deformation at position $\vec{r}$ that is projected in the direction of the spheroid center, $\vec{e}_r(\vec{r})$ is the unit vector that points radially away from the spheroid center (at position $\vec{r}$), and $\delta$ is the differentiation constant. We choose $\delta$ to be of the same size as the window size used in the PIV method for determining the deformation field.

Using the semi-affine material model as defined above, we calculate the radial uniaxial stress $\sigma(\vec{r})$ from the determined strain $\varepsilon(\vec{r})$ (see https://saenopy.readthedocs.io for Python code examples). Finally, we evaluate the local matrix stiffness

$$k(\vec{r}) = \frac{\partial \sigma(\vec{r})}{\partial \varepsilon(\vec{r})} \tag{18}$$

using numerical differentiation.

## Single-cell 3D traction force microscopy

3D traction force microscopy is conducted as explained in *Cóndor et al. (2017)*. In brief, we pipet 1.75 ml of collagen solution into a 35 mm Petri dish and let it set for 2.5 min at room temperature. Subsequently, we add 15,000 cells in another 250 µl of collagen and add this solution on top to obtain a 2 mm-thick layer of collagen. This two-layer approach prevents cells from sinking to the bottom before the gel polymerizes. After waiting for one hour to ensure the complete polymerization of the gel, 2 ml of cell culture medium are added. An additional waiting time of at least two hours before imaging ensures that cells have properly spread into a polarized shape within the collagen gel. In each independent experiment, we image a cubic volume V=(370 µm)$^3$ around up to 40 individual cells using confocal reflection microscopy (20× water dip-in objective with NA 1.0). We subsequently add cytochalasin D (20 µM), wait 30 min to ensure actin fiber depolymerization, and repeat the imaging. Based on the measured deformation fields, we obtain the cell contractility and force polarity of 90 individual A172 cells and 86 individual U87 cells from three independent experiments each.

## DNA isolation and proliferation of U87 cells

To establish a standard curve for cell number quantification, the DNA of 2000, 4000, 16000, 32000 U87 cells are extracted and quantified as described below. To measure cell proliferation during the 24 h spheroid formation process, the DNA of U87 spheroids (n = 7) grown for 24 h from 7500 cells is quantified. To measure cell proliferation during 24 h of culture in collagen, U87 spheroids (n = 7) (grown for 24 h from 7500 cells) are embedded in collagen and incubated for an additional 24 h, followed by DNA extraction. To prepare collagen-embedded spheroids for DNA extraction, single spheroids are pipetted and incubated in basal DMEM media (200 µl) containing 1x collagenase/hyaluronidase (Stem Cell Technologies) for 1 hr at 37˚C.

For DNA extraction and quantification, 250 µl of 1x cell lysis buffer (final concentration 20 mM Tris-HCL pH 7.4, 15 mM EDTA pH 8.0, 1 % SDS) is added to U87 cell or spheroids (or 50 µl of 4x cell lysis buffer is added to collagenase/hyaluronidase-treated collagen embedded spheroids). All samples are treated with RNAse A (50 µg/ml) for 30 min at 37˚C and with Proteinase K (250 µg/ml) overnight at 37˚C. The DNA is extracted with phenol/chloroform/isoamyl alcohol (Sigma), then back extracted with 100 µl TE buffer (10 mM Tris-HCL pH 7.5, 1 mM EDTA pH 8.0) and the DNA is precipitated. All DNA measurements are performed using the QuantiFluor dsDNA System kit with a Quantus Fluorometer (all Promega) according to the manufacturer's protocol.

## Code availability

The traction force microscopy method introduced in this work is implemented in the Python package jointforces, which provides an interface to the meshing software Gmsh (*Geuzaine and Remacle, 2009*) and includes particle image velocimetry functions to analyze time-lapse image data. The software is open source (under the MIT License) and is hosted on GitHub (https://github.com/christoph-mark/jointforces; *Böhringer and Mark, 2020*). For material simulations and to obtain material parameters from macrorheological measurements, jointforces uses saenopy, a Python-port of the network optimizer SAENO (*Steinwachs et al., 2016*). saenopy is open source (under the MIT License) and is hosted on GitHub (https://github.com/rgerum/saenopy; *Gerum, 2020*). The figures in this study have been created using the Python packages (*Hunter, 2007* and *Gerum, 2019*).

## Acknowledgements

This work was supported by Deutsche Forschungsgemeinschaft (DFG, German Research Foundation) grants FA-336/11–1, STR 923/6–1, TRR 225 project 326998133 (subprojects C02 and B01), the Research Training Group 1962 'Dynamic Interactions at Biological Membranes: From Single Molecules to Tissue', the Emerging Fields Initiative of the University of Erlangen-Nuremberg, the German Academic Exchange Service (DAAD) project 'Physical mechanisms in glioblastoma cell invasion', and National Institutes of Health grant HL120839. TG was supported by an Australian Government Research Training Program Scholarship and generous funding from the Petersen Foundation.

## Additional information

### Funding

| Funder | Grant reference number | Author |
|---|---|---|
| Deutsche Forschungsgemeinschaft | FA-336/11-1 | Christoph Mark<br>David Böhringer<br>Nadine Grummel<br>Ben Fabry |
| Deutsche Forschungsgemeinschaft | STR 923/6-1 | Pamela L Strissel<br>Reiner Strick |
| Deutsche Forschungsgemeinschaft | TRR 225 project 326998133 (subproject B01) | Pamela L Strissel<br>Reiner Strick |
| Deutsche Forschungsgemeinschaft | RTG 1962 | Christoph Mark |
| Deutsche Forschungsgemeinschaft | TRR 225 project 326998133 (subproject C02) | Nadine Grummel<br>Ben Fabry |

| Emerging Fields Initiative of the University of Erlangen-Nuremberg | | David Böhringer |
| German Academic Exchange Service | | Christoph Mark<br>Thomas J Grundy<br>Geraldine M O'Neill<br>Ben Fabry |
| National Institutes of Health | HL120839 | Ben Fabry |
| Australian Government Research Training Program Scholarship | | Thomas J Grundy |
| Petersen Foundation | | Thomas J Grundy |

The funders had no role in study design, data collection and interpretation, or the decision to submit the work for publication.

### Author contributions

Christoph Mark, Developed the force reconstruction method, Performed the breast cancer cell/tumoroid experiments, Performed the analyses, Wrote the paper; Thomas J Grundy, Performed the glioblastoma cell experiments, Performed the analyses, Wrote the paper; Pamela L Strissel, Established the primary breast cancer cell lines, Performed the isolation and culture of the breast tumoroids, Performed the breast cancer cell/tumoroid experiments, Wrote the paper; David Böhringer, Developed the force reconstruction method, Performed the breast cancer cell/tumoroid experiments, Performed the analyses; Nadine Grummel, Performed the breast cancer cell/tumoroid experiments; Richard Gerum, Julian Steinwachs, Developed the force reconstruction method; Carolin C Hack, Matthias W Beckmann, Performed surgery and patient care; Markus Eckstein, Performed tumor dissection and histological tumor analysis; Reiner Strick, Designed the study, Established the primary breast cancer cell lines, Performed the isolation and culture of the breast tumoroids, Wrote the paper; Geraldine M O'Neill, Ben Fabry, Designed the study and wrote the paper

### Author ORCIDs

Christoph Mark https://orcid.org/0000-0002-8612-6469

### Ethics

Human subjects: Human tissue collection was approved by the Ethics Committee of the Friedrich-Alexander University Erlangen-Nürnberg, Germany (#99_15Bc) in accordance with the World Medical Association Declaration of Helsinki. Informed consent was obtained from all patients.

### Decision letter and Author response

Decision letter https://doi.org/10.7554/eLife.51912.sa1
Author response https://doi.org/10.7554/eLife.51912.sa2

## Additional files

### Supplementary files

• Transparent reporting form

### Data availability

The source code for data analysis (deformation field estimation, force measurements) have been uploaded to Github (https://github.com/christophmark/jointforces; copy archived at https://github.com/elifesciences-publications/jointforces). All experimental time-lapse imaging data of spheroids and tumoroids used in this work are publicly available in the following Dryad repository: https://data-dryad.org/stash/share/L2Sr-Wk8eMQ6jgcYi08-4G4_Z05Tv3u6yUlmBNKymG8.

The following dataset was generated:

| Author(s) | Year | Dataset title | Dataset URL | Database and Identifier |
|---|---|---|---|---|
| Mark C, Grundy TJ, Strissel PL, Böhringer D, Grummel N, Gerum R, Steinwachs J, Hack CC, Beckmann MW, Eckstein M, Strick R, O'Neill GM, Fabry B | 2020 | Data for 'Collective forces of tumor spheroids in three-dimensional biopolymer networks' | https://datadryad.org/stash/dataset/doi:10.5061/dryad.h44j0zpf4 | Dryad Digital Repository, 10.5061/dryad.h44j0zpf4 |

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
