## [Decision Letter]

**Acceptance summary:**

In this manuscript Mark et al. described a powerful approach to extend previous work on measuring 3D traction forces in single cells in non-linear materials to multicellular tumor spheroids. Although spheroids are increasingly used for different studies, no satisfying tools existed so far to address this question in non-linear materials at this scale. Mark et al. succeed to quantify the contractility of spheroids embedded in a collagen network by combining simulations with simple bright field sectioning-free imaging. This method can now be used for other biopolymer networks.

**Decision letter after peer review:**

Thank you for submitting your article "Collective forces of tumor spheroids in three-dimensional biopolymer networks" for consideration by *eLife*. Your article has been reviewed by three peer reviewers, and the evaluation has been overseen by a Reviewing Editor and Anna Akhmanova as the Senior Editor. The following individuals involved in review of your submission have agreed to reveal their identity: Timo Betz (Reviewer #1); Pierre Nassoy (Reviewer #2).

The reviewers have discussed the reviews with one another and the Reviewing Editor has drafted this decision to help you prepare a revised submission.

Summary:

In this manuscript Mark et al. describe a powerful approach to extend previous work on measuring 3D forces in single cells in non-linear materials to multicellular tumor spheroids. Although these systems are more and more used, no satisfying tools exist to address this question in non-linear materials at this scale. They succeed in quantifying the contractility of spheroids embedded in a collagen network by combining simulations with simple bright field sectioning-free imaging.

The reviewers and myself consider that the tools to measure the contractility of organoids or tumoroids reported here would, in principle, fill a critical need for the growing community using organoids.

Essential revisions:

1) Nevertheless, the extension to collagen types other than that used in this study is not straightforward. To be more widely used, we suggest that the authors measure and list the parameters corresponding to the most commonly used collagen concentrations (for instance, for 0.5, 1,1.5, 2, 2.5, 3 µm/ml), polymerized at 37 degrees and at room temperature. Additionally, authors should also provide a clear workflow about how to get the relevant parameters for other, custom collagen gels.

2) In addition, we think that the robustness of the model should be further challenged and that different questions on data analysis and on the stiffness heterogeneity in the gel should be addressed. For instance, a spherical geometry is considered whereas the spheroids and the matrix can have more complex geometries. Furthermore, while the model fits well with the data, a validation of the magnitudes of the pressures reported by the model and when they deviate from the model should be discussed.

On Figure 1G, the deformation field is not symmetric (stronger in the lower right corner), although due to force balance the forces in the different directions should be equal. A simple explanation is that the collagen stiffness is not homogeneous in the different directions. Can the authors quantify these inhomogeneities (maybe by measuring the variance in the radial deformation fields in different directions)? Could they provide a stiffness mapping of the simulated collagen matrix during the experiments? It would be very instructive to see whether a reinforcement occurs close to the periphery of the spheroid.

3) There were some issues on the biological part of the paper. For instance, the twitching of the cells was not clearly detectable, and showing 5 different cell types with different dynamics is not very instructive if no constructive discussion is provided.

In conclusion, we thus suggest reducing the biological message and the number of cell types investigated and to rather reinforce the discussion of the model and its generalization.

---

## [Author Response]

Essential revisions:1) Nevertheless, the extension to collagen types other than that used in this study is not straightforward. To be more widely used, we suggest that the authors measure and list the parameters corresponding to the most commonly used collagen concentrations (for instance, for 0.5, 1,1.5, 2, 2.5, 3 µm/ml), polymerized at 37 degrees and at room temperature. Additionally, authors should also provide a clear workflow about how to get the relevant parameters for other, custom collagen gels.

We agree. In the revised manuscript, we list the material parameters for three different, widely used collagen concentrations (0.6mg/ml, 1.2mg/ml, 2.4mg/ml) based on our protocol. Additionally, we provide material parameters for Fibrin and Matrigel matrices.

For scientists who wish to employ polymerization protocols or use materials that have not yet been mechanically characterized, we provide detailed descriptions and ready-to-use fitting routines to obtain the material parameters from standard rheology experiments, as part of the Python package saenopy(see https://saenopy.readthedocs.io). The parameters obtained from these fitting routines can be directly inserted into the force reconstruction algorithm provided by this work. We refer to these new tools in the revised Materials and methods section.

2) In addition, we think that the robustness of the model should be further challenged and that different questions on data analysis and on the stiffness heterogeneity in the gel should be addressed. For instance, a spherical geometry is considered whereas the spheroids and the matrix can have more complex geometries. Furthermore, while the model fits well with the data, a validation of the magnitudes of the pressures reported by the model and when they deviate from the model should be discussed.

We agree and have added Figure 4G-I where we analyze how the measured deformation field (matrix deformations versus radial distance from the center of the spheroid) deviate from the predicted deformations, in particular in the case of highly elongated tumoroids (see Figure 4A in the revised manuscript). Importantly for the robustness of our method, however, the resulting error in the pressure estimate falls below 10% if we only consider matrix deformations beyond a distance of 3-4 radii. Since in our method we compute the contractile pressure as the median of the locally inferred pressure values for distances > 2 radii, the resulting error will remain substantially below 10%, and thus we can state with confidence that our method is highly robust even in the case of large deviations from spherical symmetry.

On Figure 1G, the deformation field is not symmetric (stronger in the lower right corner), although due to force balance the forces in the different directions should be equal. A simple explanation is that the collagen stiffness is not homogeneous in the different directions. Can the authors quantify these inhomogeneities (maybe by measuring the variance in the radial deformation fields in different directions)? Could they provide a stiffness mapping of the simulated collagen matrix during the experiments? It would be very instructive to see whether a reinforcement occurs close to the periphery of the spheroid.

These are important points that are now addressed in the revised manuscript. The main source of directional asymmetry in the deformation field are not so much inhomogeneities in linear collagen stiffness per se but asymmetries in the local density of cells at or near the spheroid surface that engage with and pull on the collagen matrix. Because of the pronounced stress stiffening of the collagen fibers, even relatively small local force fluctuations can thus lead to large stiffness heterogeneities and hence asymmetries in the matrix deformations – exactly as the reviewer already suspected. We followed the reviewer’s suggestion and now provide stiffness maps of the simulated collagen matrix during the experiments for an exemplary spheroid and tumoroid (Videos 5, 7).

Moreover, following the reviewer’s suggestion, we have quantified these heterogeneities (see Figure 4—figure supplement 1 and Videos 8-11) by subdividing the deformation field around spheroids and tumorids into narrow 5° angular segments. We then compute the contractile pressure for each segment. Despite the very large directional heterogeneity of the deformation field around the usually non-spherical tumorids, we find that the resulting directional variability (coefficient of variation, relative sd) in contractile pressure is only 0.16 on average. In the case of spheroids, the directional heterogeneity of the deformation field is of course considerably smaller, and the directional variability for 5° angular segments is only 0.07 on average. Thus, the pronounced asymmetry in the deformation field around some spheroids and most turmoids does not lead to substantial errors.

3) There were some issues on the biological part of the paper. For instance, the twitching of the cells was not clearly detectable, and showing 5 different cell types with different dynamics is not very instructive if no constructive discussion is provided.In conclusion, we thus suggest to reduce the biological message and the number of cell types investigated and to rather reinforce the discussion of the model and its generalization.

We now provide data recorded at 1 min intervals for which the twitching is much more clearly detectable (see revised Figure 6). We agree with the reviewer that the biological part is not the main strength and focus of this “tools and resources” study. The reason why we measure spheroids from different cell types is to show that our method can handle a large range of maximum contractile forces and a large variety of force build-up dynamics. We also compare single cell forces with collective forces. Moreover, the editor asked us to demonstrate that our method is generally applicable for spheroids from different cell types (and not just for glioblastoma spheroids as in our initial submission).